# Somatic DNA Variants in Epilepsy Surgery Brain Samples from Patients with Lesional Epilepsy

**DOI:** 10.3390/ijms26020815

**Published:** 2025-01-19

**Authors:** Jana Marie Schwarz, Lena-Luise Becker, Monika Wahle, Jessica Faßbender, Ulrich-Wilhelm Thomale, Anna Tietze, Susanne Morales-Gonzalez, Ellen Knierim, Markus Schuelke, Angela M. Kaindl

**Affiliations:** 1Department of Neuropediatrics, Charité-Universitätsmedizin Berlin, Corporate Member of Freie Universität Berlin, Humboldt-Universität Berlin, and Berlin Institute of Health, 13353 Berlin, Germany; jana-marie.schwarz@charite.de (J.M.S.); lena-luise.becker@charite.de (L.-L.B.); monika.wahle@charite.de (M.W.); jessica.fassbender@charite.de (J.F.); susanne.morales-gonzalez@charite.de (S.M.-G.); e.knierim@drk-kliniken-berlin.de (E.K.); 2Center for Chronically Sick Children, Charité-Universitätsmedizin Berlin, Corporate Member of Freie Universität Berlin, Humboldt-Universität Berlin, and Berlin Institute of Health, Augustenburger Platz 1, 13353 Berlin, Germany; 3German Epilepsy Center for Children and Adolescents, Charité-Universitätsmedizin Berlin, Corporate Member of Freie Universität Berlin, Humboldt-Universität Berlin, and Berlin Institute of Health, Augustenburger Platz 1, 13353 Berlin, Germany; 4Charité Pediatric Head and Neck Center, Charité-Universitätsmedizin Berlin, Corporate Member of Freie Universität Berlin, Humboldt-Universität Berlin, and Berlin Institute of Health, Augustenburger Platz 1, 13353 Berlin, Germany; 5German Center for Child and Adolescent Health (DZKJ), Partner Site Berlin, 13353 Berlin, Germany; 6Institute of Cell and Neurobiology, Charité-Universitätsmedizin Berlin, Corporate Member of Freie Universität Berlin, Humboldt-Universität Berlin, and Berlin Institute of Health, Charitéplatz 1, 10117 Berlin, Germany; 7Department of Pediatric Neurosurgery, Charité-Universitätsmedizin Berlin, Corporate Member of Freie Universität Berlin, Humboldt-Universität Berlin, and Berlin Institute of Health, Augustenburger Platz 1, 13353 Berlin, Germany; ulrich-wilhelm.thomale@charite.de; 8Institute of Neuroradiology, Charité-Universitätsmedizin Berlin, Corporate Member of Freie Universität Berlin, Humboldt-Universität Berlin, and Berlin Institute of Health, Augustenburger Platz 1, 13353 Berlin, Germany; anna.tietze@charite.de; 9Neurocure Clinical Research Center, Charité–Universitätsmedizin Berlin, Charitéplatz 1, 10117 Berlin, Germany

**Keywords:** structural epilepsy, genetic epilepsy, drug-resistant epilepsy, DRE, epilepsy surgery, somatic variants, WES, Mutect2

## Abstract

Epilepsy affects 50 million people worldwide and is drug-resistant in approximately one-third of cases. Even when a structural lesion is identified as the epileptogenic focus, understanding the underlying genetic causes is crucial to guide both counseling and treatment decisions. Both somatic and germline DNA variants may contribute to the lesion itself and/or influence the severity of symptoms. We therefore used whole exome sequencing (WES) to search for potentially pathogenic somatic DNA variants in brain samples from children with lesional epilepsy who underwent epilepsy surgery. WES was performed on 20 paired DNA samples extracted from both lesional brain tissue and reference tissue from the same patient, such as leukocytes or fibroblasts. The paired WES data were jointly analyzed using GATK Mutect2 to identify somatic single nucleotide variants (SNVs) or insertions/deletions (InDels), which were subsequently evaluated in silico for their disease-causing potential using MutationTaster2021. We identified known pathogenic somatic variants in five patients (25%) with variant allele frequencies (VAF) ranging from 3–35% in the genes *MTOR*, *TSC2*, *PIK3CA*, *FGFR1*, and *PIK3R1* as potential causes of cortical malformations or central nervous system (CNS) tumors. Depending on the VAF, we used different methods such as Sanger sequencing, allele-specific qPCR, or targeted ultra-deep sequencing (amplicon sequencing) to confirm the variant. In contrast to the usually straightforward confirmation of germline variants, the validation of somatic variants is more challenging because current methods have limitations in sensitivity, specificity, and cost-effectiveness. In our study, WES identified additional somatic variant candidates in additional genes with VAFs ranging from 0.7–7.0% that could not be validated by an orthogonal method. This highlights the importance of variant validation, especially for those with very low allele frequencies.

## 1. Introduction

Epilepsy is one of the most common neurological disorders, affecting approximately 50 million people affected worldwide [1]. The use of anti-seizure medication (ASM) is typically the first-line treatment for epilepsy. However, when seizures persist despite the administration of two appropriately selected ASMs, the condition is classified as “drug-resistant epilepsy” (DRE) [2]. In such cases, epilepsy surgery becomes a potential alternative, especially for structural DRE. Unfortunately, even when surgery is performed with curative intent and believed to be performed appropriately, approximately 30% of patients fail to achieve seizure freedom, sometimes with recurrences years after the initial surgery [3]. The reasons why surgery does not result in a cure in approximately one-third of these cases are the subject of active research and remain partly unclear [4,5].

Several hypotheses have been proposed for the molecular basis of DRE, including the “genetic variation hypothesis”, the “therapeutic target hypothesis”, and the “drug transporter hypothesis” [1]. It has been hypothesized that changes in the expression or function of drug-metabolizing enzymes, drug targets, or drug transporters may predispose to DRE, but this has been demonstrated in only a few examples. Such changes could be induced either directly by genetic alterations in the respective coding genes, or indirectly by the activation of inflammatory pathways, ultimately leading to secondary changes in gene expression [1,6].

In our study, we searched for pathogenic or likely pathogenic somatic variants in brain tissue excised during epilepsy surgery in children with focal structural epilepsy. For a description of the patient cohort, see Table 1. The identification of potentially pathogenic germline variants using WES of DNA extracted from blood was not the focus of this study but has been previously investigated. We depicted these results in Table 1. WES data from these previous investigations were used in the present study solely for paired data analysis in order to identify somatic variants and to exclude germline variants. In general, structural brain abnormalities can include malformations of cortical development (MCD), tumors, and hippocampal sclerosis (HS), although the latter is more commonly observed in adults. All of these abnormalities have been shown to play a role in DRE [7]. The first two conditions—(i) focal cortical dysplasia (FCD), hemimegalencephaly, and polymicrogyria as common forms of MCD, and (ii) common tumor types such as dysembryoplastic neuroepithelial tumors (DNET) and gangliogliomas (GG)—are particularly important in the pediatric population. Seizures are often the only symptom associated with these slow-growing, typically benign tumors, which are hence referred to as long-term epilepsy-associated tumors (LEATs) [8,9]. In addition, lesions acquired early in life, such as those resulting from perinatal infarction or hemorrhage, are a further common cause of lesional epilepsy in children. FCDs are classified by the International League Against Epilepsy (ILAE) into three different types based on their histopathologic appearance [10,11]: (i) FCDI is an isolated lesion with focal abnormalities of vertical (Ia), horizontal (Ib), or both vertical and horizontal (Ic) cortical lamination, (ii) FCDII is characterized by dysmorphic neurons with (IIB) or without (IIa) balloon cells, and (iii) FCDIII denotes malformed cortical lamination plus other lesions, such as hippocampal sclerosis (IIIa), tumors (IIIb), vascular malformations (IIIc), or other, e.g., post-traumatic or inflammatory lesions (IIId).

Epilepsy surgery aims to remove the epileptogenic tissue and often leads to significant improvement in seizure control, even in very young children [12,13,14,15]. However, despite surgical intervention, understanding the underlying genetic causes of these epileptogenic lesions remains critical, especially when seizures persist after surgery. Historically, the identification of epilepsy-related genes has focused on inherited germline mutations [16]. The first epilepsy genes discovered were those encoding ion channels, such as the *CHRNA4* gene [17], which encodes a nicotinic acetylcholine receptor, and the *KCNQ2* gene [18], which encodes a voltage-gated ion channel. These findings introduced the concept of monogenic “channelopathies”, in which mutations in a single ion channel gene would cause an increase in neuronal excitability and subsequently generalized epilepsy [19]. While this mechanism explained only a small fraction of cases, it provided a compelling basis for understanding the genetics of epilepsy. Subsequently, the identification of other epilepsy-associated genes beyond ion channels revealed novel pathogenic principles. Advances in sequencing technologies, particularly next-generation sequencing (NGS), have since expanded the understanding of the genetic drivers of epilepsy. As of September 2024, the latest Genomics England Panel [20] for early onset or syndromic epilepsy (v6.4) includes 857 genes known to cause epilepsy in a monogenic, Mendelian fashion. In recent years, research efforts have focused on the detection of a wider range of genetic variants, including somatic variants, which are usually not inherited. These are genetic alterations that arise postzygotically and are restricted to a specific cell type or tissue, e.g., brain tissue, and which have been identified as contributing to the pathogenesis of both MCD and tumors [21,22,23,24,25], thus representing an important cause of focal structural epilepsy.

One of the most studied genetic pathways implicated in developmental lesions is the mechanistic target of the rapamycin (mTOR) pathway. Somatic variants in genes within this pathway, such as *MTOR*, *TSC1/2*, *DEPDC5*, and *AKT3* have been frequently identified in patients with epilepsy-associated MCD [21,22,26,27,28]. LEATs are often associated with mutations in genes of the mitogen-activated protein kinase (MAPK) pathway, including *BRAF* and *FGFR1* [24,29].

The aim of the present study was to detect pathogenic somatic small variants, especially SNVs and InDels, in brain biopsy specimens that might have contributed to the (drug-resistant) epilepsy in our patients. We applied WES to DNA extracted from fresh-frozen brain tissue and from blood leukocytes or fibroblasts. Our data analysis considered mechanisms related to both drug resistance and the development of structural lesions or LEATs.

## 2. Results

### 2.1. Technical Summary

In all cases included in the study, the quantity and quality of brain tissue samples were adequate for further analysis. Histopathology was not available for two subjects. DNA extracted from brain tissue was used for WES at 100× coverage (for 15 samples) or 500× coverage (for 5 samples) (Table 1). Sequencing at a mean depth of 100× yielded an average of 172 SNVs and small InDels after Mutect2 filtering steps (Figure 1), and sequencing at a higher depth of 500× yielded an average of 11.942 SNVs and small InDels. The higher coverage was chosen during the study to increase the reliable detection of low-frequency variants.

### 2.2. Description of the Patient Cohort

The study included 20 children with structural epilepsy (Table 1). Patients underwent presurgical evaluation as previously described [30] and received epilepsy surgery between the ages of 3 months and 17 years. Patients were grouped according to the histopathology of resected tissue (if available) and/or radiologic findings: (A) inflammation, (B) MCD, (C) hippocampal sclerosis, (D) vascular malformation, (E) brain tumors, and (F) post-traumatic lesions.

### 2.3. Targeted Variant Search by Etiology, Based on Histopathologic/Radiologic Findings

We first searched for pathogenic somatic SNVs in genes known to be involved in each patient’s pathology using dedicated virtual gene panels. Depending on the group to which patients were assigned, we used the following virtual gene panels: (A) inflammation/autoinflammation panel (Appendix A), (B) MCD/MCD panel (Appendix A), (C) hippocampal sclerosis/epilepsy panel (Appendix A), (D) vascular malformation/vascular malformation panel (Appendix A), (E) brain tumor/tumorigenesis panel (Appendix A), or (F) post-traumatic lesion/epilepsy panel (Appendix A). We detected pathogenic somatic variants in genes of the mTOR pathway in three out of ten individuals with MCD (30%) and in two out of four patients with CNS tumors (50%) (for a summary see Table 2).

#### 2.3.1. *MTOR*

In brain sample 3, we identified a previously reported somatic missense variant in *MTOR* (c.6644C>T, p.S2215F, NM_004958) with a variant allele frequency (VAF) of 13% in the WES data (Appendix A). Confirmatory TAS showed a brain VAF of 7.5% (Appendix A) and DMAS qPCR of 1.6% (Appendix A). Clinically, the patient had structural DRE with up to ten focal-onset seizures per day since the age of 7 months, despite ASM with lamotrigine, lacosamide, levetiracetam, and valproate. Radiology revealed a hemimegalencephaly (HMEG) (Figure 2a), and the patient underwent a hemispherotomy at the age of 2.5 years. Histopathology revealed reactive gliosis. The patient remained seizure-free for two months post-surgery while still on ASM but was lost to further follow-up. Blood WES did not identify a pathogenic germline variant. The somatic *MTOR* variant has been previously reported as a somatic variant in brain biopsy specimens from individuals with FCDIIb [21] and HMEG. According to the additional specifications to the ACMG/AMP Sequence Variant Interpretation Guidelines [31] to be used for somatic variants [32], the variant can be classified as “pathogenic”. It is also listed in ClinVar (ID 156703), where it is classified as “pathogenic” for germline occurrences and it is a known hotspot mutation [33]. We believe that the variant identified in our patient’s brain sample is most likely associated with the formation of the HMEG as an epileptogenic lesion, as single gain-of-function variants in positive regulators of the mTOR pathway (e.g., in *PIK3CA*, *MTOR*, or *RHEB*) are sufficient to cause FCDII or HMEG [28,34]. This finding again emphasizes the central role of mTOR dysregulation in the pathogenesis of focal MCD such as HMEG.

#### 2.3.2. *PIK3CA*

A previously reported somatic missense variant in *PIK3CA* (c.1624G>A, p.E542K, NM_006218) was detected in brain sample 4, with a VAF of 29% in WES data (Appendix A) and confirmed by Sanger sequencing (Appendix A). Clinically, the patient presented with macrocephaly and DRE, with up to five focal-onset seizures of altered consciousness per day since the age of 4 months. Other features included syndactyly and polydactyly and the patient was diagnosed with Greig cephalopolysyndactyly syndrome (GCPS, OMIM #175700 [35]). A radiologic examination revealed right-sided focal hemimegalencephaly and an extensive migration disorder (Figure 2b). EEG showed unilateral nonconvulsive status epilepticus. Medication with oxcarbazepine, lamotrigine, levetiracetam, and clonazepam was ineffective. The patient underwent a hemispherotomy at the age of 1.9 years and has been seizure-free to date, recently without taking any ASM. Histologic analysis of a brain biopsy, which included medullary and ependymal tissue but not the cortex, showed predominantly reactive tissue changes. WES with DNA extracted from blood identified a paternally inherited heterozygous pathogenic germline deletion of 10 bp in *GLI3* (ClinVar variation ID 972686). Of note, the patient’s father and grandfather were also diagnosed with GCPS, but had no history of seizures. This *PIK3CA* variant has been previously reported as a somatic variant in brain biopsy specimens from patients with HMEG [21,36,37]. According to the additional specifications of the ACMG/AMP Sequence Variant Interpretation Guidelines [31] to be used for somatic variants [32], the variant can be classified as “pathogenic”. It is also listed in ClinVar (ID 31944) and marked as “pathogenic” in germline cases. Greig cephalopolysyndactyly syndrome is typically characterized by frontal bossing, polydactyly and variable syndactyly, peculiar cranial shape, and hypertelorism with variable expressivity. The *GLI3* variant in our patient explains the cephalopolysyndactyly phenotype but not necessarily the epileptogenic brain malformation. However, since it has been shown that somatic variants in *PIK3CA* lead to the formation of HMEG, and the p.E542K variant identified in our patient is located in a mutational hotspot of the gene [33], we believe that the somatic *PIK3CA* variant detected in our patient causes the epileptogenic brain lesion.

#### 2.3.3. *TSC2*

In brain sample 5, a previously reported nonsense variant in *TSC2* (c.3442C>T, p.Q1148*, NM_000548) was detected with a VAF of 3% in the WES data (Appendix A), 2.5% in the confirmatory TAS (Appendix A) and 1.7% in DMAS-qPCR (Appendix A). Interestingly, the same *TSC2* variant was also identified in fibroblasts from a skin punch biopsy with a higher VAF of 6% in DMAS-qPCR (Appendix A) and 7.5% in TAS (Appendix A). Since both skin fibroblasts and CNS neurons are derived from the same germ cell layer, the ectoderm, we hypothesize that the somatic variant occurred in a common ectodermal progenitor cell. Clinically, the patient had DRE with up to five focal aware seizures per day since the age of 2 years. cMRI revealed focal cortical dysplasia (FCD) in the right precuneal area (Figure 2c). Interictal EEG showed normal background activity with epileptiform discharges in the right occipital region. Seizure control was not achieved with treatment using levetiracetam and oxcarbazepine. The patient underwent epilepsy surgery at the age of 3.5 years with complete resection of the FCD. Histopathology was consistent with FCDIIb. Postoperatively, the patient experienced four additional seizures with the previously known semiology but is now seizure-free for more than two years on lamotrigine monotherapy. Blood WES revealed a heterozygous probable pathogenic nonsense variant in *ATP1A2* (ClinVar variation ID 2498438), which was, however, maternally inherited. Variants in *ATP1A2* are associated with various forms of migraine, and with epilepsy and developmental and epileptic encephalopathy type 98 (DEE98) [38,39,40]. Homozygous truncating mutations in *ATP1A2* have been associated with early lethal hydrops fetalis, arthrogryposis, microcephaly, and polymicrogyria [41], although this seems to be a very rare case. Of note, the patient’s mother suffers from migraine, but not from epilepsy. The somatic *TSC2* variant has been previously reported as a germline variant in individuals with tuberous sclerosis (TSC) [42,43,44] and is listed in ClinVar (ID 50087) as “pathogenic” for germline occurrences. Current knowledge suggests that brain lesion formation typically requires a “double hit” mechanism involving both a germline and a somatic loss-of-function variant in mTOR pathway repressors, such as *TSC2* [10]. To investigate a potential “second hit” in mTOR repressors in our patient, in addition to the somatic *TSC2* variant, we reanalyzed leukocyte the WES data for pathogenic SNVs, small InDels, or larger deletions/CNVs but we did not identify any additional relevant pathogenic germline variant in mTOR pathway genes, except for the *ATP1A2* variant. Given reports of pathogenic somatic heterozygous variants in *TSC1* or *TSC2* causing FCDII without a second hit [45,46], we hypothesize a similar mechanism in this patient, possibly resulting from haploinsufficiency.

#### 2.3.4. *FGFR1* & *PIK3R1*

In brain sample 15, we detected previously reported pathogenic somatic variants in *FGFR1* and *PIK3R1*. The *FGFR1* variant (c.1966A>G, p.K656E, NM_023110) had a VAF of 36% in the WES data (Appendix A) and was confirmed by Sanger sequencing (Appendix A). The *PIK3R1* variant (c.1690A>G, p.N564D, NM_181523) had a VAF of 33% in brain WES (Appendix A), also confirmed by Sanger sequencing (Appendix A). Clinically, the patient had a single bilateral tonic-clonic seizure of unknown origin at the age of 16 years. cMRI revealed a lesion in the right occipital gyrus, suspected to be a low-grade glioneuronal tumor (Figure 2d). The interictal EEG was normal. The patient underwent a lesionectomy shortly after the seizure, based on a suspected tumor diagnosis, despite epilepsy not being officially classified as drug-refractory. He is seizure-free to date. Histopathologic analysis revealed a low-grade glioneuronal tumor (LGGNT). In genome-wide methylation analysis, the DNA methylation profile could not be assigned to any known methylation class, but the highest classifier score was obtained for the rosette-forming glioneuronal tumor (RGNT) class (score slightly below the cutoff for classification). WES on fibroblast DNA was performed for joint calling of somatic variants in the brain sample, without additional evaluation of germline variants. The *FGFR1* variant is listed in ClinVar (ID 224897) as “pathogenic” or “likely pathogenic” for germline occurrences. It is a known gain-of-function hotspot mutation [47] and has been reported as a somatic variant in RGNTs and other cancers [24]. The *PIK3R1* variant is also listed in ClinVar (ID 376261) as “pathogenic” or “likely pathogenic” for somatic origin and has been frequently detected in vascular malformations and various cancers [48].

#### 2.3.5. *FGFR1* & *NF1*

In brain sample 16, we identified the same pathogenic somatic variant in the *FGFR1* gene (c.1966A>G, p.K656E, NM_023110) as in brain sample 15. WES revealed a VAF of 24% (Appendix A), the variant was also confirmed by Sanger sequencing (Appendix A). In addition, we detected a pathogenic variant in the *NF1* gene (c.2824delA, p.S942Afs*, NM_001042492, ClinVar ID 834401) with a VAF of 4.5% in the WES data, although this variant could not be confirmed by TAS (Appendix A). Clinically, the patient had epilepsy characterized by focal-onset seizures with progression to generalized tonic-clonic seizures beginning at the age of eight years. These seizures persisted despite treatment with levetiracetam. Cranial MRI showed a right temporal lesion, most likely consistent with a dysembryoplastic neuroepithelial tumor (DNET) (Figure 2e). Interictal EEG showed epileptiform discharges in the right temporal region, and ictal EEG confirmed seizure activity in the same area. Due to progressive tumor growth, the patient underwent lesionectomy at the age of 13.9 years. He remained seizure-free for 14 months after surgery. No further follow-up data were available. Histopathologic analysis was consistent with a diagnosis of DNET or RGNT, WHO grade I. WES on fibroblast DNA was performed for joint calling of somatic variants in the brain sample, without additional evaluation of germline variants.

### 2.4. Search for Variants in Drug Absorption, Distribution, Metabolism, and Excretion (ADME) Genes

To identify pathogenic somatic variants in genes associated with drug resistance, we used a previously published [49,50] panel of genes involved in drug absorption, distribution, metabolism, and excretion (ADME) to select relevant variants (Appendix A). In total, seven somatic variants predicted to be pathogenic were identified in brain WES from two patients. After visual inspection in IGV and further evaluation, two of these variants, a variant in *METAP1* (c.1037C>T, p.A346V, NM_015143) with VAF 0.8% in brain WES of patient 18 with CNS tumor and a variant in *TRPV4* (c.1796C>T, p.T599M, NM_021625) with VAF 0,72% in brain WES of patient 10 with TSC, qualified for confirmatory TAS. However, neither variant was confirmed (Appendix A).

### 2.5. Search for Variants in Novel Genes Guided by the Patient Phenotype

We also aimed to identify pathogenic somatic variants in genes not previously associated with any of the etiologies listed in Figure 1. To this end, we applied a filtering strategy based on suspected gene function or involved pathways, guided by the patient’s specific phenotype. Filtering can be done by using the appropriate function in MutationDistiller [51] (filter for GO, WikiPathways, or Reactome terms). This identified additional brain somatic variants that were considered potentially pathogenic but were not confirmed by TAS (Appendix A). These included a nonsense variant in *DAB1* (c.1000C>T, p.Q334*, NM_001365792) with a VAF of 2.7% in the brain WES dataset of patient 13 with complex malformations of cortical development (MCD) and hippocampal sclerosis, a missense variant in *PAK6* (c.1916T>A, p.L639Q, ENST00000560346) with a VAF of 1.0% in the brain WES data from patient 14 with unilateral venous cavernous malformation, and a missense variant in *APC2* (c.1940G>A, p.G647D, ENST00000233607.2) with VAF of 1.0% was found in brain WES of patient 11 with FCDIIb. As mentioned above, none of the variants were confirmed by TAS.

## 3. Discussion

The aim of this study was to further deepen our understanding of the genetic basis of lesional DRE by applying WES to DNA from brain biopsy specimens of affected individuals. Our focus was to identify pathogenic somatic variants associated with the formation of specific lesions, such as MCD and CNS tumors, as well as variants that may contribute to drug resistance. Using both histopathologic and radiologic findings, we stratified patients into six etiologic groups, allowing for targeted study using virtual gene panels relevant to each pathology. This approach allowed focused analysis of genetic alterations associated with specific structural etiologies and revealed known pathogenic somatic SNVs in established lesional epilepsy or tumorigenesis genes in 30% of patients with MCD and 50% of patients with CNS tumors, respectively.

We confirmed somatic variants in the PI3K-AKT-mTOR pathway in patients with MCD, specifically in the *MTOR*, *PIK3CA*, and *TSC2* genes. In all cases presented here, the findings provided further molecular understanding of the patients’ phenotypes, which had not been fully explained by other diagnostic methods. Our findings are consistent with previous studies demonstrating the involvement of postzygotically acquired variants in genes of this pathway in the formation of FCDII and HMEG [52,53]. However, some studies have reported even higher diagnostic rates within the MCD subgroup. The lower diagnostic rate observed in our study may be due to the fact that the optimization of sample collection and pre-analytic handling for both neuropathologic examination and DNA extraction, according to published guidelines [54], was implemented only partway through our study. As a result, it cannot be confirmed consistently that the brain biopsy samples used for DNA extraction were indeed taken from the most affected areas of the lesion.

To date, there is no clear genotype–phenotype correlation between the severity of MCD and the VAF for a specific somatic variant. However, a recent study in mice examined differences and similarities in the effects of somatic variants in either inhibitors (*Depdc5*, *Tsc1*, *Pten*) or activators (*Rheb*, *Mtor*) of the mTOR pathway on neuronal morphology, membrane excitability, and excitatory synaptic transmission [55]. The authors found that both the activation of the activators *Rheb* or *Mtor* or inactivation of the inhibitors *Depdc5*, *Tsc1*, or *Pten*—all of which boost mTORC1 activity—cause similar increases in neuronal soma size and mispositioned neurons. However, they differently affected excitatory synaptic transmission in a gene-specific manner: *Tsc1* knock-out neurons showed an increase in the frequency of spontaneous excitatory postsynaptic currents (sEPSCs) without a change in their amplitude. In contrast, neurons with activating mutations in *Rheb*, *Mtor*, and *Pten* showed an increased amplitude of sEPSCs but no change in their frequency. The underlying mechanisms for these differences remain unclear, but given the complexity of the mTOR pathway, additional signaling pathways may be involved. Identification of these additional pathways may have practical implications for more targeted therapeutic strategies.

In our study, pathogenic variants in *FGFR1* and *PIK3R1* were identified in two patients with neuroepithelial tumors, with VAFs ranging from 24–36%. These higher VAFs contrast with the lower VAFs of 3–29% observed in epilepsy lesions without tumor components. This disparity likely reflects the selective growth advantage conferred by somatic variants within tumor cells, highlighting the differences in mutational clonality and selective pressure between tumor-associated and non-tumor-associated epileptic lesions.

Given the difficulty in distinguishing LEAT entities by radiologic and histopathologic methods alone, insights into their genetic biomarkers are valuable and increasingly being applied. In the two cases presented here, molecular genetic findings further supported the histopathologic diagnoses. In brain sample 15, we identified somatic variants in *FGFR1* and *PIK3R1*, strongly supporting the diagnosis of RGNT, as variants in these genes are part of the established molecular profile for RGNTs [24,48,56]. Histopathologic analysis suggested a low-grade neuroepithelial tumor without further specification, while genome-wide methylation analysis, although not an exact match to any known methylation class, was most consistent with RGNT. The molecular genetic findings offer additional confirmation, reinforcing the histopathologic and methylation-based indications. In brain sample 16, a pathogenic somatic variant in *FGFR1* was identified, complementing the histopathologic findings suggestive of DNET, which is characterized by somatic *FGFR1* variants [29].

Overall, we identified pathogenic somatic variants in 50% of CNS tumors. Other studies have reported high diagnostic yields of up to 100% in LEATs, especially when somatic CNV analysis was included. A recent study further highlighted the role of somatic structural variations (SVs) in pediatric brain tumors [57]. The lower diagnostic yield observed in the CNS tumor subgroup in our study may be due to the fact that we focused our analysis only on SNVs and small indels, and did not include CNV and SV analysis.

We have also investigated pathogenic variants in genes not previously associated with epilepsy lesions and DRE and have identified several candidates by WES. However, confirmation of variants, especially those with very low VAF, has proven challenging. Variant validation primarily aims to distinguish false positives from bona fide somatic variants using an orthogonal method. Different methods are available (e.g., Sanger sequencing, TAS, ddPCR (digital droplet PCR), DMAS-qPCR, and others). Key considerations for our choice of a specific method comprised VAF, experimental effort and cost, and the number of variants to be validated. We used Sanger sequencing for higher VAF variants due to its cost-effectiveness, and commercial TAS for lower VAF variants due to its higher sensitivity, despite its higher cost. In our study, none of the very low-VAF variants could be validated by TAS. This limitation in detecting low-VAF somatic variants in epilepsy lesions using alternative methods has been mentioned by other authors as well [22] and has been attributed to sequencing errors in WES or artifacts during variant calling. In addition, it is possible that targeted amplicon sequencing as a validation method lacks sensitivity to detect very low abundance variants.

We first attempted allele-specific double mismatch qPCR (DMAS-qPCR) for variant confirmation, a technique known for its sensitivity, with detection thresholds for variant frequencies as low as 1% or even less [58]. DMAS-qPCR is widely used for SNP genotyping and mutation detection in cancer research and is most effective when controls for known hotspot mutations are available. It has also been used for confirmation of somatic variants in brain malformation studies [32]. In our case, we wanted to validate novel mutations without available controls. Relying on differences in C_t_-values between affected DNA from brain biopsy specimens and reference DNA from fibroblasts or leukocytes from the same patient proved reduced reliability in our setting, as reflected by differences in VAF between DMAS-qPCR and WES/TAS in the two cases where we used both methods. In addition, primer design and PCR optimization posed significant challenges, as the 3’ ends of the oligonucleotide primers had to be positioned at the variant site, limiting our primer design options. For these reasons, DMAS-qPCR was unsuitable for large-scale variant validation, leading us to adopt TAS as an alternative method. Although amplicon sequencing is theoretically capable of detecting very low VAF, it still presents significant technical challenges. The benefit of high coverage comes at the cost of nearly identical duplicate reads, which add limited information and are susceptible to amplification bias and PCR error propagation, even when a high-fidelity polymerase is used in the initial PCR steps for amplicon generation. To address these issues, it is increasingly recommended to introduce unique molecular identifiers (UMIs) as barcodes prior to the amplification step. This approach allows monitoring of amplification imbalances, elimination of random errors, and reliable identification of true variants at very low VAFs [59]. It should also be noted that although Illumina sequencing generates data with very low error rates (around 0.1 to 0.5%), making it reliable for most purposes, even these low error rates hinder the accurate detection of somatic variants with VAFs around or below 0.5%. Despite the challenges of variant confirmation, we confirmed pathogenic somatic variants in 25% of our patients. The diagnostic yield of our study is in line with previous publications reporting overall diagnostic yields between 10–56% or even higher for specific subgroups such as HMEG, FCDIIb, or LEATs [21,22,23,60].

## 4. Materials and Methods

### 4.1. Patient Cohort

We included 20 children with lesional epilepsy who were treated at the German Epilepsy Center for Children and Adolescents at the Charité in Berlin, and who eventually underwent epilepsy surgery. Drug resistance was defined according to the consensus proposal of the ILAE Commission as “failure of adequate trials of two tolerated, appropriately chosen and used antiepileptic drug regimens” [2]. Patients were phenotyped by pediatric neurologists and all patients underwent additional diagnostic procedures including EEG and cranial MRI. In addition, WES and, in some cases, microarray analysis of blood-derived DNA was performed to address putative underlying germline variants and genetic rearrangements as genetic causes or predisposing factors for epilepsy, the importance of which has been outlined previously [57]. When blood WES was not available, we performed WES using DNA extracted from skin fibroblasts as the reference tissue for somatic variant calling. Patients were grouped according to histopathology of resected tissue (if available) and/or radiological findings: (A) inflammation (2 patients), (B) malformations of cortical development (10 patients), (C) hippocampal sclerosis (1 patient), (D) vascular malformations (1 patient), (E) brain tumors (4 patients), and (F) post-traumatic lesions (2 patients).

### 4.2. Sample Processing and DNA Extraction

As part of the study, specimens were taken from the site of lesional alteration during epilepsy surgery for DNA extraction and histopathologic examination. Of note, genomic DNA was not extracted from the exact same biopsy specimen as used for histopathology. Genomic DNA was extracted from fresh-frozen brain tissue using the phenol-chloroform isolation method to obtain high molecular weight genomic DNA with fragment sizes of above 100 kb. Genomic DNA from blood samples was extracted using the salt extraction method according to standard protocols. Genomic DNA from fibroblasts was extracted using the NucleoSpin^®^ Tissue Kit from Macherey–Nagel (Macherey-Nagel, Düren, Germany) according to the manufacturer’s instructions.

### 4.3. Whole Exome Sequencing

WES was performed on DNA isolated from freshly frozen bulk brain tissue and peripheral blood leukocytes or from cultured skin fibroblasts as described above. Sequencing was performed by commercial sequencing service providers. For blood samples, exonic sequences and flanking intronic regions were captured using Agilent SureSelectXT (Agilent, Santa Clara, CA, USA), Twist Human Core Exome, Twist Human Comprehensive Exome + Mitochondrial Genome (Twist Bioscience, South San Francisco, CA, USA), or company-specific in-solution hybridization techniques not further specified and sequenced on Illumina machines (NextSeq, NovaSeq or HiSeq, Illumina, San Diego, CA, USA) yielding 100 bp or 150 bp paired-end reads. For brain and fibroblast samples, exonic sequences and flanking intronic regions were captured using BGI’s proprietary DNBSeq library and sequenced on a BGISEQ-500 machine, yielding 100 bp paired-end reads (BGI, Shenzhen, China). The coverage achieved was >100× on average for blood and fibroblast samples. For the brain samples, we initially targeted at least 100× coverage on average and later increased this to at least 500× to achieve better coverage of low-VAF variants. The resulting unaligned reads were further processed in-house as described below.

#### 4.3.1. Bioinformatic Processing of WES Data and Variant Calling

For all samples, FASTQ reads were aligned to the human_g1k_v37.fasta genomic reference using BWA-MEM2 v0.7.17-r1188 [61]. SNVs and small indels were then called following the Genome Analysis ToolKit (GATK) best practices for germline (if applicable) and somatic variant calling using GATK-4.4 *HaplotypeCaller* [62] (for germline variants) and GATK-4.3.0.0 *Mutect2* [63] (for somatic variants). The search for potentially pathogenic germline variants was not the focus of this study but was previously investigated, with the results added in Table 1. In the present study, blood WES data in the form of .bam files from these prior investigations were used exclusively for paired data analysis to identify somatic variants, as detailed below. In cases where no blood WES was available or no search for pathogenic germline variants intended, we performed WES on DNA from skin fibroblasts to serve as the reference tissue for somatic variant calling only. For somatic variant detection, WES data on DNA from both brain tissue and a reference tissue were analyzed as a pair in Mutect2’s matched-normal mode. This approach allows differentiation between likely germline mutations and brain-specific somatic mutations not present in the reference tissue, achieved by the tool’s built-in algorithm, which also uses a Bayesian somatic likelihood model to calculate the log odds of alleles being somatic variants vs. sequencing errors. We enabled the Mutect2 option to exclude soft-clipped bases from variant calling, as these had led to numerous likely false positives during the optimization of the data analysis protocol.

Public GATK resources were used for a Panel of Normals (PoN) (https://gatk.broadinstitute.org/hc/en-us/articles/360035890631-Panel-of-Normals-PON, accessed on 22 December 2022) to capture recurrent technical artifacts and to mark common germline variants along with their allele frequencies (af-only-gnomad.raw.sites.grch37.vcf.gz from https://storage.googleapis.com/gatk-best-practices/somatic-b37/af-only-gnomad.raw.sites.vcf, accessed on 14 November 2022). To identify high-quality somatic variants, we followed previously published recommendations [64]. Resulting variants were filtered with GATK-4.3.0.0 FilterMutectCalls and selected with SelectVariants, requiring the filter flag PASS. As an additional hard filter, alternative alleles were required to be supported by at least 4 reads. The remaining variants were analyzed and further filtered using MutationTaster2021 (version of 2021, accessed on several dates between December 2022 and December 2024) [65] and MutationDistiller (version of 2019, accessed on several dates between December 2022 and December 2024) [51], excluding common polymorphisms found in large-scale sequencing projects such as the 1000 Genomes Project [66], ExAC [67], or gnomAD [68]. Nonsense, missense, or splice site variants predicted to be disease-causing were further prioritized. All samples were screened for variants in genes potentially associated with drug resistance. For this purpose, we used a virtual gene panel (Appendix A) that lists genes involved in drug absorption, distribution, metabolism, and excretion (ADME). This gene panel has been previously used and published by other research groups involved in pharmacogenetic studies of ASM [49,50].

Variants were also prioritized according to the individual patient phenotype, e.g., for genes known to be involved in MCD (Appendix A), in tumorigenesis (Appendix A), and in vascular malformations (Appendix A). For patients with post-traumatic lesions, a broad panel of all known epilepsy genes according to the Genomics England Panel App was used (Appendix A). To extend our screening beyond known genes, we further prioritized variants based on putative gene function using Gene Ontology (GO), WikiPathways, and Reactome terms via the corresponding feature in MutationDistiller. Candidate variants were visually inspected using the Integrative Genomic Viewer (IGV) to sort out read-end artifacts and putative false positives in clusters of low-quality bases indicative of sequencing artifacts. CNVpytor-1.3.1 [69] was used for CNV analysis, when necessary.

#### 4.3.2. Variant Confirmation

Shortlisted somatic variants in DNA isolated from brain tissue were confirmed by an alternative method depending on the VAF. For variants with a VAF above 20%, we used PCR followed by automated Sanger sequencing [70] using the BigDye^®^ v3.1 Terminator protocol (Applied Biosystems, Thermo Fisher Scientific, Waltham, MA, USA) on the ABI 3500 Genetic Analyzer to confirm the presence of the variants (Appendix A for custom primers). For variants with a VAF below 20%, we used double-mismatch allele-specific quantitative polymerase chain reaction (DMAS-qPCR) [58] and/or ultra-deep targeted amplicon sequencing (TAS). For DMAS-qPCR, we designed allele-specific primers using batchprimer3 (https://wheat.pw.usda.gov/demos/BatchPrimer3/, accessed on 9 August 2023) (Appendix A for custom primers) according to published guidelines [58] and calculated the likely VAF based on the ∆∆C_t_ method (∆∆C_t_ = ∆C_t_ (wild-type allele) − ∆C_t_ (variant allele)) [71]. DMAS qPCR was later abandoned due to its difficult primer design, comparatively labor-intensive setup procedure, and partly inconclusive results.

For amplicon sequencing, we used Primer3Plus (https://www.primer3plus.com/, accessed on 24 April 2024) to design oligonucleotide primers that flank the putative variant in a fragment of 150–400 bp, with the variant ideally located between 50–80 bp on the forward strand (Appendix A for custom primers). Amplicon-EZ sequencing was then performed by a commercial service provider (Genewiz^®^/Azenta Life Sciences, Leipzig, Germany). We found it most effective to run PCR reactions in triplicates using 100 ng of template genomic DNA per 50 µL PCR reaction volume and to purify the triplicates on a single PCR purification column (Monarch^®^ Spin PCR & DNA Cleanup Kit, #T1130S, New England Biolabs, Ipswich, MA, USA) to achieve the recommended amount of DNA (500 ng DNA with a concentration of at least 20 ng/µL). In the case of unspecific PCR products, the desired band was excised from the gel and purified using a gel extraction kit (Monarch^®^ DNA Gel Extraction Kit, New England Biolabs, #T1020L). The amount of DNA was measured using a Qubit^TM^ 3.0 fluorometer and a Qubit^TM^ dsDNA HS assay kit (#Q32851, Invitrogen^TM^, Thermo Fisher Scientific, Waltham, MA, USA). A high-fidelity polymerase (Q5^®^ High-Fidelity DNA Polymerase, #M0491S, New England Biolabs, Ipswich, MA, USA) was used.

#### 4.3.3. Bioinformatic Processing of TAS Data

The FASTQ reads were trimmed using cutadapt-4.9 [69] to remove low-quality read ends, quality filtered using the fastq filter for an average PHRED score of at least 35 (https://github.com/LUMC/fastq-filter, accessed on 31 July 2024), and then aligned to the human G1Kv37 genomic reference sequence using BWA-MEM v0.7.17. Reads were visualized using the Integrative Genomic Viewer (IGV) to inspect the position of each variant.

## 5. Conclusions

In conclusion, our study elucidates the genetic basis of lesional epilepsy by analyzing somatic mutations in brain tissue samples. By integrating WES with radiologic and histopathologic data, we identified brain pathogenic somatic variants in the following, already well-established lesional epilepsy genes of the PI3K-AKT-mTOR axis: *MTOR* (in patient 3 with HMEG), *PIK3CA* (in patient 4 with focal megalencephaly), and *TSC2* (in patient 5 with FCD), highlighting their role in focal cortical dysplasia and in other epilepsy-related malformations. Moreover, we identified pathogenic somatic variants in the following genes associated with CNS tumors: *FGFR1* (in patient 15 with LGGNT and patient 16 with DNET) and *PIK3R1* (in patient 15 with LGGNT).

Identification of the molecular etiology of an epilepsy lesion primarily offers benefits to patients and affected families. These include knowledge about the cause and natural course of their disease when known somatic variants are detected. In addition, counseling regarding the risk of recurrence is relevant, since somatic variants are not passed on to offspring. Understanding pathogenic somatic variants can further clarify differences in clinical presentation in cases where both a parent and their child carry the same germline variant. In such cases, the additional somatic variant may explain the variation in symptom severity. Furthermore, although we could not demonstrate it, pathogenic somatic variants could, in principle, contribute to drug resistance. In our study, the identification of somatic variants had no direct therapeutic implications because all patients had undergone epilepsy surgery, and epileptogenic lesions were completely resected in the individuals in whom the pathogenic somatic variants were identified. However, in the case of incomplete resection or when somatic variants are detected in biopsy samples, the identification of pathogenic somatic variants may be relevant for more personalized epilepsy treatments, including the selection of ASM, pathway-specific treatments such as mTOR inhibitors, and therapeutic strategies for associated tumor management.

## Figures and Tables

**Figure 1 ijms-26-00815-f001:**
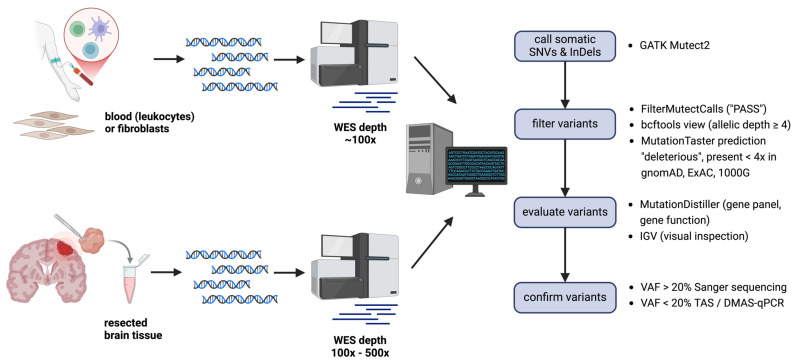
Single steps performed to identify pathogenic somatic variants in brain biopsy specimens. Mutect2: GATK tool for somatic variant calling; FilterMutectCalls: GATK tool to filter real Mutect2 calls from errors, artifacts, germline variants, and more, probable somatic variants are given the flag “PASS”; bcftools: a tool to manipulate vcf files. It was used to keep only calls with an allelic depth of ≥4); MutationTaster: online variant prediction tool to distinguish pathogenic from benign variants, this tool was also used to filter out common variants from gnomAD, ExAC, and 1000G; MutationDistiller: online variant prioritization tool, used to apply different virtual gene panels taking into account the clinical phenotype of the patients and to prioritize variants based on assumed gene function (pathway-based); IGV: genome browser to display WES data. Abbreviations: DMAS-qPCR, double-mismatch allele-specific quantitative polymerase chain reaction; InDel, insertion and deletion, SNV, single nucleotide variant; TAS, targeted amplicon sequencing; VAF, variant allele frequency; WES, whole exome sequencing. The image was created with BioRender.

**Figure 2 ijms-26-00815-f002:**
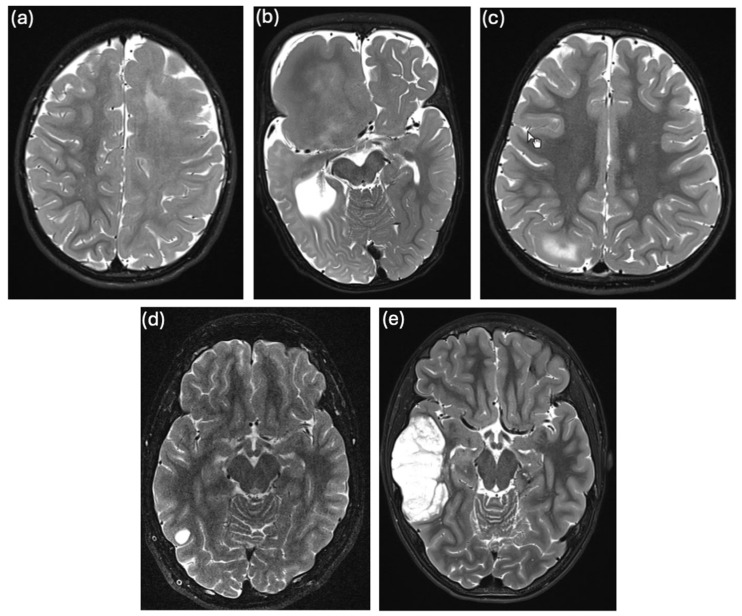
T_2_-weighted cMRI images of individuals in whom we detected pathogenic somatic variants: (**a**) individual 3 with left-sided HMEG, (**b**) individual 4 with right-sided focal megalencephaly, (**c**) individual 4 with FCDIIb in the right-sided precuneal area, (**d**) individual 15 with a small right-sided, temporo-occipital lesion suspicious of LGGNT, and (**e**) individual 16 with a right-sided, temporal DNET.

**Table 1 ijms-26-00815-t001:** List of all subjects included in the study with accompanying additional information, allocated to the following etiologic groups: (A) inflammation, (B) malformations of cortical development, (C) hippocampal sclerosis, (D) vascular malformation, (E) brain tumors, and (F) post-traumatic lesions. Investigated patient material: (^1^) subdural (not cortical) sampling; (^2^) cortical biopsy, (^3^) HC biopsy. Abbreviations: DNET, dysembryoplastic neuroepithelial tumor; FCD, focal cortical dysplasia; GNT, glioneuronal tumor; HC, hippocampus; het, heterozygous; HMEG, hemimegalencephaly; HS, hippocampal sclerosis; ICA, internal carotid artery; LGGNT, low-grade glioneuronal tumor; MCA, middle cerebral artery; MCD, malformation of cortical development; MVNT, multinodular vacuolating neuronal tumor; nd, not determined; PMG, polymicrogyria; RE, Rasmussen encephalitis; RGNT, rosette-forming glioneuronal tumor; TSC, tuberous sclerosis; WES, whole exome sequencing. * For detailed information on somatic variants, see Table 2.

Brain Sample/Individual	Group	Age at Surgery [Years]	Radiologic/Histologic Diagnosis	Pathogenic Germline Variant in Blood (Gene/Region)	Pathogenic Somatic Variant in Brain (Gene) *	Coverage Brain WES	Number of Somatic Variants in Brain WES After Mutect2 Filtering
1	A	8.8	cMRI: RE; histology: consistent with RE	no	no	100×	53
2	A	14.5	cMRI: RE; histology: reactive gliosis, consistent with RE	no	no	100×	253
3	B	2.5	cMRI: HMEG; histology: reactive gliosis	no	*MTOR*	100×	127
4	B	1.9	cMRI: focal megalencephaly; histology: reactive changes	*Gli3* (NM_000168.6), c.4430_4439del (het), p.S1477fs	*PIK3CA*	100×	145
5	B	3.5	cMRI: FCD; histology: FCDIIb	*ATP1A2* (NM000702.4), c.2827C>T (het), p.Q943*	*TSC2*	100×	217
6	B	1.9	cMRI: FCD; histology: FCDIIb	nd	no	100×	250
7	B	10.5	cMRI: PMG, hemiatrophy; no histology	Del2p15	no	100×	181
8	B	0.5	cMRI: TSC; histology: TSC	*TSC2* (NM_000548.5), c.4606C>T (het), p.Q1536* and Del17q12	no	100×	149
9	B	16.9	cMRI: FCD; histology: FCDIIa	no	no	100×	118
10	B	5.3	cMRI: TSC; histology: TSC	*TSC1* (NM_000368.5), c.2257dup (het), p.S753Kfs*8	no	500×	9058
11	B	0.3	cMRI: FCD; histology: FCDIIb	no	no	500×	11,241
12	B	4.5	cMRI: complex MCD, PMG; histology: connective tissue ^1^	no	no	500×	10,700
13	C	8.5	cMRI: FCD, HC atrophy; histology: focal cortical gliosis ^2^, HS type I ^3^	no	no	100×	171
14	D	2.5	cMRI: unilateral cavernous venous malformation; histology: cavernous dilatation of vessels, consistent with venous angioma	no	no	500×	6723
15	E	16.7	cMRI: LGGNT; histology: DNET or RGNT	nd	*FGFR1*, *PIK3R1*	100×	120
16	E	13.9	cMRI: DNET; histology: DNET	nd	*FGFR1*	100×	340
17	E	9.0	cMRI: DNET, MVNT or other GNT; histology: LGGNT	nd	no	100×	158
18	E	9.2	cMRI: glial tumor; histology: glial tumor	no	no	500×	21,988
19	F	14.7	cMRI: partial MCA infarction; histology: acute hypoxic-ischemic infarction	no	no	100×	120
20	F	15.0	cMRI: ICA occlusion; no histology	no	no	100×	176

**Table 2 ijms-26-00815-t002:** List of pathogenic somatic variants identified in brain samples that were confirmed by at least one orthogonal method. Description of etiologic groups: (B) malformations of cortical development, (E) brain tumors. fb, fibroblasts, DMAS, double-mismatch allele-specific, Sanger = Sanger sequencing, TAS, targeted amplicon sequencing, VAF, variant allele frequency, WES, whole exome sequencing.

Brain Sample/Individual	Group	Gene/Variant	VAF Brain WES	VAF in Blood and in fb	Validation Method (VAF Brain if Applicable)
3	B	*MTOR* (NM_004958) c.6644C>T, p.S2215F,	13%	blood 0%	TAS (7.5%) DMAS qPCR (1.6%)
4	B	*PIK3CA* (NM_006218) c.1624G>A, p.E542K,	29%	blood 0%	Sanger
5	B	*TSC2* (NM_000548) c.3442C>T, p.Q1148*,	3%	blood 0% fb 6–7.5%	TAS (2.5%) DMAS qPCR (1.7%)
15	E	*FGFR1* (NM_023110) c.1966A>G, p.K656E, *PIK3R1* (NM_181523) c.1690A>G, p.N564D	36% 33%	fb 0% fb 0%	Sanger Sanger
16	E	*FGFR1* (NM_023110) c.1966A>G, p.K656E	24%	fb 0%	Sanger

## Data Availability

The data presented in this study or subsets thereof are available on reasonable request from the corresponding author due to strict data protection rules for patient-derived genetic data. Human genomic data cannot be deposited in public repositories without consent, which we do not have.

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
