# Peer review of "Somatic DNA Variants in Epilepsy Surgery Brain Samples from Patients with Lesional Epilepsy"

_ijms, 2025, doi:10.3390/ijms26020815_

Round 1

Reviewer 1 Report

Comments and Suggestions for Authors

The paper, "Somatic DNA variants in epilepsy surgery brain samples from patients with lesional epilepsy," is well-designed and written. The authors attempted to identify DNA variants that may contribute to the lesions and/or lead to severe symptoms of epilepsy. Somatic DNA of brain samples from children with lesional epilepsy have been used for whole exome sequencing (WES). The authors identified  MTOR, TSC2, PIK3CA, FGFR1, and PIK3R1 genes as potential causes of cortical malformations and tumors. 

I have a major criticism of this work:

1) It is not clear, how the authors distinguished somatic and germline DNA variants that may contribute to the pathology.

2)The authors need to add a conclusion to this paper that describes all key findings in detail.

3) Have the authors added raw sequencing data to NCBI databases?

Minor:

1) Figure 1 has a pure resolution. The authors need to increase the resolution and improve the figure legend.

Author Response

The paper, "Somatic DNA variants in epilepsy surgery brain samples from patients with lesional epilepsy," is well-designed and written. The authors attempted to identify DNA variants that may contribute to the lesions and/or lead to severe symptoms of epilepsy. Somatic DNA of brain samples from children with lesional epilepsy have been used for whole exome sequencing (WES). The authors identified  MTOR, TSC2, PIK3CA, FGFR1, and PIK3R1 genes as potential causes of cortical malformations and tumors. 

I have a major criticism of this work:

[1] It is not clear, how the authors distinguished somatic and germline DNA variants that may contribute to the pathology.

Answer:    In this study, we focused exclusively on identifying pathogenic somatic variants. However, for 18 of the 20 patients, germline variant analysis using whole exome sequencing (WES) had beenb previously performed following standard guidelines. To clarify this, we have revised the manuscript and included the following sentence (p.2, l.78): "The identification of potentially pathogenic germline variants using WES of DNA extracted from blood was not the focus of this study but has been previously investigated, with the results added in Table 1. Blood WES data from these previous investigations were used in the present study solely for paired data analysis to identify somatic variants and exclude germline variants." Additionally, we have incorporated relevant details into the methods section to further clarify this point: ”The search for potentially pathogenic germline variants was not the focus of this study but was previously investigated, with the results added in Table 1. In the present study, blood WES data in the form of .bam files from these prior investigations were used exclusively for paired data analysis to identify somatic variants, as detailed below. In cases where no blood WES was available or no search for pathogenic germline variants intended, we performed WES on DNA from skin fibroblasts to serve as reference tissue for somatic variant calling only.” The distinction between somatic variants and germline variants (the latter are not emitted by Mutect2) is achieved by the tools’ build-in algorithm (this information has also been added to the manuscript). In cases where we previously identified pathogenic germline variants as well as pathogenic somatic variants, the relevance of both, germline and somatic variants in terms of pathogenicity, is discussed in detail in the Results section (see section 2.3.2 and 2.3.3).

[2] The authors need to add a conclusion to this paper that describes all key findings in detail.

Answer:    We have added a dedicated section heading to the conclusion of our results, at the end of the main text of our manuscript. In this section, we begin with reiterating the five genes in which somatic variants were detected, along with the corresponding patient pathologies (the latter were newly added), to provide a concise conclusion. Moreover, we draw overall conclusions on the importance of the study of somatic variants in terms of genetic counseling and therapeutic strategies.

[3] Have the authors added raw sequencing data to NCBI databases?

Answer:    Due to confidentiality in order to protect sensitive private genomic data, we are not allowed to upload the complete raw data sets without specific consent, which we do not have. In the data availability lines at the end of the article wie state: “Data Availability Statement: The data presented in this study or subsets thereof are available on reasonable request from the corresponding author due to strict data protection rules for patient-derived genetic data. Human genomic data cannot be deposited in public repositories without consent, which we do not have.”

Minor:

[4] Figure 1 has a pure resolution. The authors need to increase the resolution and improve the figure legend.

Answer:    Figure 1 has been added at a higher resolution (600 dpi). The figure legend has been improved.

Reviewer 2 Report

Comments and Suggestions for Authors

The article explores the role of somatic DNA variants in lesional epilepsy through whole exome sequencing of brain samples from pediatric patients undergoing epilepsy surgery. By analyzing genetic alterations in lesional tissue and comparing them to reference tissue, the study identified pathogenic variants, particularly in the PI3K-AKT-mTOR pathway, and their association with malformations of cortical development and central nervous system tumors. The findings underline the complexity of epilepsy's molecular basis and the challenges in validating low-variant allele frequencies. The work emphasizes the significance of integrating genetic, radiologic, and histopathologic data for better understanding of the etiology and potential therapeutic implications in drug-resistant epilepsy cases. Following are my major concerns regarding this article:

1.    The study utilized brain tissue with varying coverage depths (100x and 500x), which could influence the detection of low-frequency variants. A call for action is to standardize the sequencing coverage across all samples to ensure uniform variant detection sensitivity and improve the reliability of findings.

2.   The reliance on different validation methods, including Sanger sequencing, TAS, and DMAS-qPCR, introduces inconsistencies. To address this, the authors should apply a single robust and sensitive validation method, such as UMI-based deep sequencing, to confirm all low-frequency variants.

3.   The study mentions incomplete optimization of sample collection for both histopathologic examination and DNA extraction. Please ensure synchronized and optimized sample processing protocols to enhance genotype-phenotype correlation accuracy.

4.  The lower diagnostic yield for MCD and CNS tumors compared to other studies suggests potential methodological or analytic gaps. The authors should re-evaluate data analysis pipelines and incorporate advanced computational tools for more effective variant prioritization.

5.   The filtering process relies heavily on the GATK pipeline, but further measures, such as using multiple variant callers and stricter thresholds for read support, should be implemented to minimize false positives and improve variant calling reliability.

6.   Stratification based on histopathology and radiologic findings, while useful, may not capture the genetic heterogeneity within subgroups. The authors should consider additional stratification using clinical phenotypes and family history to refine genetic analyses.

7.    The study’s focus on known ADME gene panels may exclude novel pathways contributing to drug resistance. I strongly suggest performing unbiased genome-wide association studies or transcriptomic analyses to identify additional mechanisms.

Author Response

The article explores the role of somatic DNA variants in lesional epilepsy through whole exome sequencing of brain samples from pediatric patients undergoing epilepsy surgery. By analyzing genetic alterations in lesional tissue and comparing them to reference tissue, the study identified pathogenic variants, particularly in the PI3K-AKT-mTOR pathway, and their association with malformations of cortical development and central nervous system tumors. The findings underline the complexity of epilepsy's molecular basis and the challenges in validating low-variant allele frequencies. The work emphasizes the significance of integrating genetic, radiologic, and histopathologic data for better understanding of the etiology and potential therapeutic implications in drug-resistant epilepsy cases. Following are my major concerns regarding this article:

[1] The study utilized brain tissue with varying coverage depths (100x and 500x), which could influence the detection of low-frequency variants. A call for action is to standardize the sequencing coverage across all samples to ensure uniform variant detection sensitivity and improve the reliability of findings.

Answer:    Our study spanned several years. We agree with the reviewer that higher coverage is desirable. During the course of the study, we already increased the coverage for brain WES in five samples. This was made possible by advancements in sequencing technologies, and decreased sequencing costs over time. However, due to the limited material available from brain biopsies and financial constraints, we were unable to re-sequence the initial 100x coverage brain samples with higher coverage. Despite this, we consider our findings to be reliable, which is also illustrated by the fact that we identified a low frequency variant (VAF 1.7-2.5%) in the 100x coverage data.

[2] The reliance on different validation methods, including Sanger sequencing, TAS, and DMAS-qPCR, introduces inconsistencies. To address this, the authors should apply a single robust and sensitive validation method, such as UMI-based deep sequencing, to confirm all low-frequency variants.

Answer:    In the setting of our study, variant validation primarily aims to distinguish false positives from bona fide somatic variants using an orthogonal method. This method must differ from the initial variant detection method, but does not need to be same for all the variants. Indeed, numerous other studies have used different methods for variant validation, since each method has its limitations. The choice of method should be carefully tailored to the variant type and study scope. Key considerations in our case included variant allele frequency (VAF), experimental effort and cost, and the number of variants to be validated. We used Sanger sequencing for higher VAF variants due to its cost-effectiveness, and commercial TAS for lower VAF variants due to its higher sensitivity, despite its higher cost. Both methods have been used alongside each other in previous studies (e.g. Lai et al. Brain 2022, https://doi.org/10.1093/brain/awac117 and Pelorosso et al. Human Molecular Genetics 2019, https://doi.org/10.1093/hmg/ddz194). DMAS-qPCR (which was also applied by Lai et al. 2022) was discontinued during our study, with TAS replacing it for all low VAF variants. This ensures consistent and uniform validation while acknowledging that different variants may require and justify different approaches. While UMI-based TAS principally is desirable, it is not yet routinely available for custom DNA samples at Genewiz/Azenta. This might however change in the future, allowing for improved validation of very low-frequency variants. To clarify our choice for different validation methods, we added the following sentence to the manuscript: Variant validation primarily aims to distinguish false positives from bona fide somatic variants using an orthogonal method. Different methods are available (e.g. Sanger sequencing, TAS, ddPCR, DMAS-qPCR and others). Key considerations for the choice of method in our case included variant allele frequency (VAF), experimental effort and cost, and the number of variants to be validated. We used Sanger sequencing for higher VAF variants due to its cost-effectiveness, and commercial TAS for lower VAF variants due to its higher sensitivity, despite its higher cost.”

[3] The study mentions incomplete optimization of sample collection for both histopathologic examination and DNA extraction. Please ensure synchronized and optimized sample processing protocols to enhance genotype-phenotype correlation accuracy.

Answer:    In our manuscript, we explained that the sample collection process was optimized over the course of the study, which spanned several years. Ethical considerations obviously prevent us from re-performing brain surgeries or biopsies. As such, we cannot ensure optimized sample processing for samples already collected during earlier surgeries.

[4] The lower diagnostic yield for MCD and CNS tumors compared to other studies suggests potential methodological or analytic gaps. The authors should re-evaluate data analysis pipelines and incorporate advanced computational tools for more effective variant prioritization.

Answer:    As outlined in our answer to the previous point [3], delayed sample collection procedure might account for lower diagnostic rate in the subgroup of MCDs and this unfortunately cannot be overcome retrospectively. The lower diagnostic rate in CNS tumors might be due to the fact that we focused our study on SNVs and InDels, by design of the study. Since we followed published best practice guidelines, we are confident that the choice of our data analysis pipeline is effective for the scope of our study (to identify somatic SNVs and InDels).

[5] The filtering process relies heavily on the GATK pipeline, but further measures, such as using multiple variant callers and stricter thresholds for read support, should be implemented to minimize false positives and improve variant calling reliability.

Answer:    While a multiple-caller approach is sometimes recommended, numerous small and large studies published in high impact journals rely on the GATK best practices guidelines and Mutect2 as a single caller (e.g., Lai et al. Brain 2022, https://doi.org/10.1093/brain/awac117; Lopez-Rivera et al. Brain 2023, https://doi.org/10.1093/brain/awac376; Zhang et al. Epilepsia 2020, https://doi.org/10.1111/epi.16481), as did we. We agree with the reviewer that additional thresholds may be necessary, particularly for high-coverage data. Therefore, we already applied an additional hard filter of a minimum alternative allele depth of 4 (see Figure 1 and Methods’ section 5.3.1).

[6] Stratification based on histopathology and radiologic findings, while useful, may not capture the genetic heterogeneity within subgroups. The authors should consider additional stratification using clinical phenotypes and family history to refine genetic analyses.

Answer:    We totally agree with the reviewer that the clinical phenotype should be considered to refine genetic analyses. He might have missed that we already describe different strategies for variant prioritization: in section 2.3, we describe our targeted variant search by etiology, based on histopathologic / radiologic findings; in section 2.5, we describe our filtering strategy based on suspected gene function or involved pathways, guided by the patient's specific phenotype. We now added this information to the subheading of section 2.4, to make this more clear (“2.5. Search for variants in novel genes guided by the patient phenotype”).

[7] The study’s focus on known ADME gene panels may exclude novel pathways contributing to drug resistance. I strongly suggest performing unbiased genome-wide association studies or transcriptomic analyses to identify additional mechanisms.

Answer:    As already outlined in our answer to the previously raised point [6], we do not focus on ADME genes, but this was only one strategy of variant prioritization amongst others. The perfomance of GWAS studies requires large patient and control cohorts with members counting in the thousands which will surly neither ethically (brain surgey/biopsy in controls (!)) nor pracitally feasable. The biopsy material obtained for our study was only sufficient to extract DNA for WES, but not additionally RNA for transcriptome analysis so that we will not be able to realize this reviewer’s suggestion.

Round 2

Reviewer 1 Report

Comments and Suggestions for Authors

I recommend to accept it in its present form.